# XLNet-Caps: Personality Classification from Textual Posts

**Ying Wang** [1,2,3] ![ID], **Jiazhuang Zheng** [1,2], **Qing Li** [1,2] ![ID], **Chenglong Wang** [1,2] ![ID], **Hanyun Zhang** [1,2] ![ID] and **Jibing Gong** [1,2,3,*] ![ID]

1   The Key Laboratory for Computer Virtual Technology and System Integration of Hebei Province, Qinhuangdao 066004, China; wangying@ysu.edu.cn (Y.W.); zhengjiazhuang169@stumail.ysu.edu.cn (J.Z.); liqing1@stumail.ysu.edu.cn (Q.L.); chenglongwang@stumail.ysu.edu.cn (C.W.); zhanghanyun@stumail.ysu.edu.cn (H.Z.)
2   School of Information Science and Engineering, Yanshan University, Qinhuangdao 066004, China
3   Key Laboratory for Software Engineering of Hebei Province, Yanshan University, Qinhuangdao 066004, China
*   Correspondence: gongjibing@ysu.edu.cn

**Abstract:** Personality characteristics represent the behavioral characteristics of a class of people. Social networking sites have a multitude of users, and the text messages generated by them convey a person's feelings, thoughts, and emotions at a particular time. These social texts indeed record the long-term psychological activities of users, which can be used for research on personality recognition. However, most of the existing deep learning models for multi-label text classification consider long-distance semantics or sequential semantics, but problems such as non-continuous semantics are rarely studied. This paper proposed a deep learning framework that combined XLNet and the capsule network for personality classification (XLNet-Caps) from textual posts. Our personality classification was based on the Big Five personality theory and used the text information generated by the same user at different times. First, we used the XLNet model to extract the emotional features from the text information at each time point, and then, the extracted features were passed through the capsule network to extract the personality features further. Experimental results showed that our model can effectively classify personality and achieve the lowest average prediction error.

**Keywords:** machine learning; XLNet; capsule; Big Five model; deep learning; NLP

## 1. Introduction

Personality encompasses a person's behaviors, emotions, motivations, and thought patterns. In a few words, our personality traits determine what we might say and do. Personality differences among individuals are an important research direction in psychology. In the field of computers, we can, through the calculation analysis of user behavior, obtain results to achieve the purpose of predicting the user's personality, as well as to quantify individual differences and take advantage of these to predict the demands of the individual users, to achieve the aim of providing better personalized services to them [1]. In marketing, more suitable products can be recommended according to personality characteristics. In the field of psychology, for a psychologist, if one can understand the state of a patient's personality, this will be beneficial to the patient's targeted treatment. For individuals, if patients with depression can identify their state early, this may significantly reduce suicidal behavior. For society, to some extent, such an analysis can help suppress crime.

With the rapid development of network technology, network ideology has become more intricate, and social media have become a prevalent tool for social activities. Through various social media and information communication platforms, a growing number of Internet users have published their views. Social media contain much self-disclosure of personal information and emotional content; netizens' expression of their individual ego orientation in the network space and its variability are far greater than in real society. In real society, there is a general phenomenon, due to the external constraints of the social environment, that most individuals' expression is a reflection of their natural attributes.

However, in cyberspace, this typical expression deviates to a certain extent, and extreme expression becomes possible. By deeply analyzing the characteristics of network information content, including social network media platforms, through natural language processing, the emotional tendencies of Internet users, and the deep personality characteristics of Internet users in combination with the psychological analysis model, problems in the fields of social science and social practice can be solved well.

Psychologists believe that personality is a person's stable attitude towards reality and his/her corresponding habitual behavior. From Allport's pioneering work, to Cutter's 16 Root Traits, to the proposal of the Big Five personality traits, a fundamental assumption has been made throughout, and that is the lexical hypothesis: important aspects of human life are given words to describe; if something is fundamental and ubiquitous, in all languages, it will be given more words to describe. Therefore, discovering personality traits from vocabulary has become a significant approach to personality research.

Traditional mainstream approaches are mostly manual extractions of lexical and grammatical features in the text. These characteristics can distinguish character traits. These features are used to select a suitable classifier to effectively classify personality based on text content. The concrete implementation of the method uses feature vectors as the input based on the SVM classifier, but there are some drawbacks. This method of feature selection requires considerable time. Due to the short-text noise problem, the effect is also not ideal.

In addition, in recent years, deep learning-based neural networks and distributed representations have been influential in sentence/document modeling. They have achieved superior performance in natural language processing (NLP) applications, such as text-based sentiment analysis and opinion mining. Notably, these NLP applications seem to be similar to our personality recognition tasks, as they all involve mining user-attribute text through techniques such as text and feature representation.

Common technologies are CNNs and RNNs. CNNs have a weight-sharing network structure to reduce the complexity, thus the parameters that need to be trained, of the network model, resulting in them being more similar to biological neural networks. The same weights can keep the filter from being affected by the influence of the position signal when detecting the signal features, and the same weights give the trained model a better generalization ability. The pooling operation can reduce the spatial resolution of the network, thus eliminating the minor deviation and distortion of the signal. However, CNNs cannot model the changes in time series and cannot analyze the overall logical sequence of the input information; they are also prone to gradient dissipation. RNNs have overcome the problem of CNNs, which do not analyze the overall logic of the relations of the input sequence. The key is that the hidden state of network retains the previous output information that was used as the input to the network. The depth of the model is in the time dimension, which can be seen as sequence modeling. Its disadvantage is that too many parameters are needed, causing the gradient dissipation or gradient explosion problem, and the model does not have the ability to learn the characteristics.

In the past natural language processing algorithms, a natural language was usually processed as a word vector, with the disadvantage being that if the word or sentence is mapped into a vector, the semantics cannot be fully utilized. Space-insensitive methods are inevitably limited by rich text structures (such as the storage of word locations, semantic information, and the grammatical structure), are difficult to effectively encode, and lack text expression ability. A capsule network can improve the above drawbacks. The capsule network uses a vector to represent a feature instead of a scalar representing a feature, making the feature expression richer and have complete semantics.

This paper proposed a deep learning-based method to judge the personality of social network users' text. XLNet-Caps was used as the language feature extraction model. Taking into account the emotional difference of each user at different times, the emotional color of the published social text is different [2–4]. This model was divided into two levels: XLNet was used to extract the features of the text published by the same user at different

times, and as the upgraded BERT model, XLNet replaced the AE model with the AR model to solve the side-effects of the mask. It took the dual-flow attention mechanism, carried out a deeper study of the context, and pre-processed the text better; the text was treated using XLNet different times, adopting capsules for further feature extraction, which could be performed automatically by the model and reflected most of the key language features, including words, sentences, or subjects, of the user's psychological characteristics. Then, the model extracted the feature information using the neural network to classify and judge the user's personality. This model is not just a one-sided study of a person's psychological characteristics, but also a comprehensive analysis of a person's personality characteristics by analyzing a large number of texts published by a person at different times.

This model was mainly based on Tappes' Big Five Theory of Personality [5,6]. Other studies have shown that the Big Five model will not cause differences in the results due to differences in the language, nor in the testing and analysis methods [5,7]. Table 1 introduces the various idiosyncratic factors of the Big Five personality model. Emotional polarities are mainly divided into five categories, including extroversion, agreeableness, responsibility, emotional stability, and openness.

**Table 1.** Big Five model.

| Trait Factors | Extroverted (Extroverted) | Neuroticism (Emotional Stability) | Openness to Experience (Openness) | Pleasant (Easy-Going) | Serious (Cautious) |
|---|---|---|---|---|---|
| Description | socially decisive, passionate, lively, courageous, optimistic | disquiet, animosity, dismay, self-conscious, impulsivity, vulnerability | imagination, aesthetics, emotional enrichment, difference, intelligence | trust, frankness, altruism, compliance, modesty, empathy | competence, organization, dedication, accomplishment, self-discipline, caution |

The main contributions of this paper are as follows:

(1) In this paper, a text feature extraction method based on the capsule model—XLNet-Caps—was proposed. By optimizing the embedded XLNet layer and capsule layer features, the model could extract the features of the text at a deeper level.
(2) The features learned from our neural network were more effective than the ten other baseline models' features, and our identification method outperformed all the other baselines with the lowest prediction error.

Organization: The rest of this article is organized as follows. The Section 2 briefly introduces the previous scholars' related work and research. The Section 3 discusses the proposed methodology. The Section 4 introduces the implementation details, and the experimental results and analysis are given. Finally, in the Section 5, the conclusions and the direction for future work are given.

## 2. Related Work

Early methods of personality classification were mainly based on manually defined rules. With the innovation and development of deep learning technology, methods based on neural networks have gradually become mainstream. On this basis, many researchers use language knowledge and text classification techniques to better classify personality.

### 2.1. Traditional Personality Classification

Many methods of traditional personality classification are focused on feature engineering. Then, the carefully designed functions are provided to the machine learning method in a supervised learning environment. Therefore, the performance of personality classification depends mainly on the choice of text feature representation. For feature engineering, the most commonly used function is the bag-of-words function. In addition, there are some more complex designs, such as the fastText model [8], which uses a word

sequence as the input. The words in the sequence become a feature vector, and then, the feature vector is mapped to the middle layer through a linear transformation. The middle layer is mapped to the label, and finally, the probability that the word sequence belongs to different categories is output. In addition to supervised learning, Reference [9] introduced an unsupervised method and used emotional words/phrases extracted from syntactic patterns to determine the polarity of the document. In terms of features, different types of representations are used in sentiment analysis, including bag-of-words representation, word co-occurrence, and sentence context [10]. Although functionally practical, feature engineering is labor-intensive, and it is impossible to extract and organize multi-scale identifying information from the data.

### 2.2. Deep Learning for Personality Classification

Since proposing a simple and effective method to learn distributed representations of words and phrases [11], neural network-based models have shown great success in many NLP tasks. Many models have been applied to text classification, and these text classifications are used in research on personality classification, including the following: The TextCNN model [12] first maps the text to a vector and then uses multiple filters to capture the local semantics of the text information. Then, it uses maximum pooling to capture the most important features and, finally, inputs these features into the fully connected layer to obtain the probability distribution of the label. The DPCNN model [13] can obtain the best performance by increasing the network depth without greatly increasing the computational overhead. Multi-task learning [14] is also gradually being applied in this field. Furthermore, the RCNN model uses a recursive structure to capture contextual information and uses a convolutional neural network to construct a representation of the text. The TextGCN model, which constructs a large heterogeneous text graph containing word nodes and document nodes, explicitly models the global word–word co-occurrence information, so it regards the text classification problem as a node classification problem. B. Felbo et al. [15] proposed a hybrid neural network combining Bi-LSTM and an attention mechanism, which performed well in the emotion recognition of emoticons, and similarly, it performed well in text classification tasks. Convolutional-gated recurrent neural networks [16] and hierarchical attention networks [17] are constantly emerging from the literature for the task of document-level text classification.

Since the capsule network was proposed, many studies have been carried out on this basis. W. Zhao et al. [18] proposed a text classification model based on the capsule network, improved the dynamic routing proposed by Sabour et al., exploring the use of the dynamic routing capsule network for text classification, and proposed three strategies to stabilize dynamic routing to reduce the distribution of noise capsules. Y. Wang et al. [19] proposed RNN-capsule. The BERT pre-training model [20] overcomes the problem of static word vectors not being able to solve polysemous words. Its dynamic word vectors based on language models have achieved the best performance in multiple tasks in natural language processing. As a result, it has opened a new chapter in many research fields in NLP. After BERT first introduced the two-stage model, more and more research has appeared along these lines. Z. Yang et al. [21] proposed the XLNet model. The XLNet model adopts the AR model to replace the AE model. It solves the problems caused by the mask model, adopts a dual-flow attention mechanism, and introduces the Transformer XL architecture. It has surpassed BERT's performance on more than 20 tasks.

## 3. Methodology

### 3.1. Problem Formulation

The main task is to judge the user's personality through the user's social text, that is personality classification. In reality, due to the users' different backgrounds, tones, and other circumstances when sending a social text, there may be similar social texts with different meanings. It is difficult for humans to distinguish the true meaning of this kind of special social text, but this type accounts only for a small proportion of all social

text. Therefore, we assumed that these special types do not exist. Consider a document $D = \{s_1, s_2, ..., s_n\}$, where $s_i$ denotes the social text of the $i$-th user and $n$ denotes the number of all users. Consider a set of tags $Y = \{y_1, y_2, ..., y_m\}$, where $y_j$ denotes the $j$-th personality and $m$ denotes the number of all personalities. The purpose of the paper was to obtain a personality prediction set $Y'_i = \{y'_1, y'_2, ..., y'_{m'}\}$ by predicting the personality of each given $s_i$, where $y'_j$ denotes the $j$-th personality of the $i$-th user and $y'_j \in Y$ and $m'$ denotes the number of all personalities of the $i$-th user and $m' \leq m$. In the final analysis, this task is in essence a multi-label multi-classification problem.

In this section, we introduce the XLNet-Caps deep learning model in detail. As shown in Figure 1, the model was divided into two levels. For the text published by the same user at different times, XLNet was used for feature extraction, and the features of the text in different time periods were extracted by XLNet. Capsules were used for further feature extraction, which means the key language features that best reflected the psychological characteristics could be automatically extracted through the model, including words, sentences, or topics, and then, the feature information was classified using neural networks to determine the user's personality. In general, our method included four stages: (1) text modeling based on time series; (2) XLNet preprocessing; (3) further extraction of features using capsules; (4) XLNet-Caps.

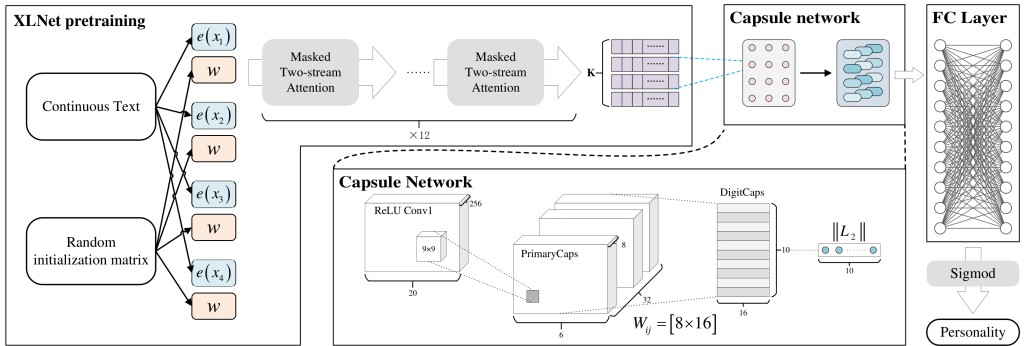

**Figure 1.** XLNet-Caps model.

### 3.2. Text Modeling Based on Time Series

The mental state of a person is not static. Therefore, the social texts sent by each person at different times can reflect their personalities from different perspectives. From this point of view, the social text sent by a person could be considered in the time dimension so that richer personality features of this person could be obtained by the model. Given a document $D$, let $D_i = \{m_1, m_2, m_3, ..., m_n\}$, where $D_i$ denotes the social text set sent by the $i$-th person and $m_j$ denotes the $j$-th message.

First, we performed preprocessing data operations, including tokenization, lemmatization, and removing stop words. Then, we converted each social text set into a sequence of a group of sentences over time [22]. Next, we used an $M \times N$ matrix to represent each text set of the user, where $M$ denotes the number of messages and $N$ denotes the number of words in each message. Note that we limited the number of messages to $M$ and the number of words contained in each message to $N$, which could speed up the training process and promote generalization. Finally, we used word embedding to map words into a meaningful vector space because this could obtain the semantic features of the text more completely and accurately. According to this idea, we could obtain an $M \times N \times D$ three-dimensional matrix, where $D$ denotes the dimension of the word vector number.

### 3.3. XLNet Pre-Processing

The advantage of pre-training is that there is no need to use a large corpus for training in specific scenarios, which saves time and is more efficient. The generalization ability of the pre-training model was strong. XLNet introduces a permutation language model. Compared with the traditional AR model, this model does not perform modeling in

order, but maximizes the expected log-likelihood of the factorization order of all possible sequences. This method not only retains the advantages of the traditional AR model, but also captures bidirectional contextual information. Given a sequence $x$ of length $T$, there are $T!$ different orders to perform a valid autoregressive factorization. The permutation language modeling objective of XLNet can be expressed as Equation (1):

$$\max_{\theta} \quad \mathbb{E}_{\mathbf{z} \sim \mathcal{Z}_T} \left[ \sum_{t=1}^{T} \log p_{\theta}(x_{z_t} \mid \mathbf{x}_{\mathbf{z}_{<t}}) \right] \tag{1}$$

where $Z_T$ denotes the set of all possible permutations of the length-$T$ index sequence $[1, 2, ..., T]$, $z_t$ denotes the $t$-th element, $z_{<t}$ denotes the first $t-1$ elements of a permutation $z \in Z_T$, and $p_{\theta}(\cdot)$ denotes the likelihood function.

XLNet introduces a two-stream self-attention mechanism to obtain the bidirectional content information of the current position without introducing a mask. The content representation $h_{\theta}(x_{z_{<t}})$ serves a similar role as the standard states in Transformer, which encode both the context and $x_{z_t}$ itself, and the content representation can be abbreviated as $h_{z_t}$. The query representation $g_{\theta}(x_{z_{<t}}, z_t)$ only has access to the contextual information $x_{z_{<t}}$ and the position $z_t$, but not the content $x_{z_t}$, as discussed above, and it can be abbreviated as $g_{z_t}$. For each self-attention layer $m = 1, ..., M$, the two streams of representation are updated as Equations (2) and (3):

$$g_{z_t}^{(m)} \leftarrow \text{ Attention } \left( Q = g_{z_t}^{(m-1)}, KV = \mathbf{h}_{z_{<t}}^{(m-1)}; \theta \right) \tag{2}$$

$$h_{z_t}^{(m)} \leftarrow \text{ Attention } \left( Q = h_{z_t}^{(m-1)}, KV = \mathbf{h}_{z_{<t}}^{(m-1)}; \theta \right) \tag{3}$$

where Q, K, and V denote the query, key, and value in the attention operation. Because of the replacement problem, the convergence speed of the permutation language model is slow. To improve the convergence speed, XLNet only predicts the last tokens in a factorization order. First, $z$ is decomposed into a non-target subsequence $z_{\leq c}$ and a target subsequence $z_{>c}$, where $c$ is the cutting point. Then, Equation (4) is used to maximize the log-likelihood of the target subsequence conditioned on the non-target subsequence.

$$\max_{\theta} \quad \mathbb{E}_{\mathbf{z} \sim \mathcal{Z}_T} \left[ \log p_{\theta}(\mathbf{x}_{\mathbf{z}_{>c}} \mid \mathbf{x}_{\mathbf{z}_{\leq c}}) \right] = \mathbb{E}_{\mathbf{z} \sim \mathcal{Z}_T} \left[ \sum_{t=c+1}^{|\mathbf{z}|} \log p_{\theta}(x_{z_t} \mid \mathbf{x}_{\mathbf{z}_{<t}}) \right] \tag{4}$$

Select the longest context in the sequence given the current factorization order $z$. For the tokens that are not selected, there is no need to calculate their query representation to save time and space. The ordinary Transformer has the longest sequence of hyperparameters to control its length. This method will cause some information to be lost when dealing with particularly long sequences. XLNet introduces Transformer-XL relative position coding and segment loop coding to resolve the dependence on ultra-long sequences. Suppose there are two segments taken from a long sequence $s$, i.e., $\tilde{x} = s_{1:T}$ and $x = s_{T+1:2T}$. Let $\tilde{z}$ and $z$ be permutations of $[1, ..., T]$ and $[T + 1, .., 2T]$, respectively. After processing the first segment, cache the obtained content representations $\tilde{h}^{(m)}$ for each self-attention layer m. For the next segment $x$, the attention update with memory is performed as Equation (5):

$$h_{z_t}^{(m)} \leftarrow \text{Attn} \left( Q = h_{z_t}^{(m-1)}, KV = \left[ \tilde{\mathbf{h}}^{(m-1)}, \mathbf{h}_{\mathbf{z}_{\leq t}}^{(m-1)} \right]; \theta \right) \tag{5}$$

where $[\cdot, \cdot]$ denotes concatenation along the sequence dimension and $Attn(\cdot)$ denotes the attention operation. The position encoding is only related to the actual positions in the original sequence. In our model framework, the output of XLNet is a feature matrix, which is used as the input to the capsule network for subsequent calculation. The composition of the XLNet model is shown in Figure 2:

- The input of XLNet is the word embedding $e(x)$ and an initialization matrix $W$. Then, the content representation and query representation are updated by several masked two-stream attention layers;
- After all calculations of the attention layers are complete, the query representation is used as the final output to capture the personality traits.

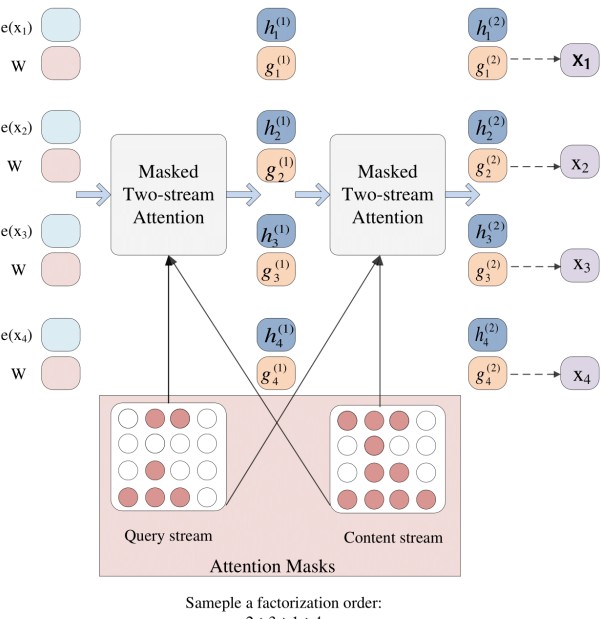

**Figure 2.** XLNet model.

The capsule network was designed for the task of image classification. However, it can be partially operated during data preprocessing or word embedding, so the capsule network can handle the task of natural language processing. Compared to CNNs, the capsule network does not lose spatial features due to the pooling layer; in other words, the capsule network is more sensitive to the position of words in sentences, so a more accurate feature matrix can be obtained. Capsule networks use position encoding technology and dynamic routing technology to replace the pooling layer in the CNN to obtain the spatial features. The capsule network can be generally considered to be composed of a ReLU convolution layer, a primary capsule layer, and a digit capsule layer [18,23], as shown in Figure 3.

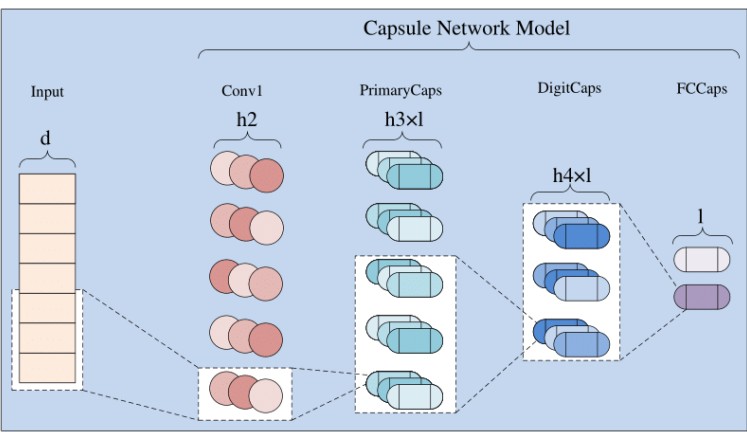

**Figure 3.** Capsule model.

### 3.4. Capsule Networks

Among these layers, the ReLU convolution layer can be simply understood as a traditional convolution layer with ReLU activation. The function of this layer is to extract the features as the input to the primary capsule layer. The function of the primary capsule layer is to convert the features extracted by the input of the ReLU convolution layer. The corresponding capsule parameters are calculated as the input of the digit capsule layer. Unlike traditional neural networks, the primary capsule layer does not piece together the instantiated parts to generate a whole, but uses the whole to obtain partial information. From the perspective of graphics operation, this process is similar to the inverse rendering process. The digit capsule layer is used to encode spatial information and conduct the final classification. This layer uses dynamic routing algorithms to optimize the final result. In order to optimize the parameters more effectively, a normalization operation has been proposed named the squash function [24,25], as shown in Equation (6). The pseudocode of the dynamic routing algorithm is provided in Algorithm 1.

$$\text{squash}(s) \ = \ \frac{\|s\|^2}{1 + \|s\|^2} \frac{s}{\|s\|} \tag{6}$$

---

**Algorithm 1** Routing algorithm.

---

1: **procedure** *Routing*$(\hat{u}_{j|i}, r, l)$
2:     for all capsules $i$ in layer $l$ and capsules $j$ in layer: $(l+1) : b_{ij} \leftarrow 0$.
3:     **for** i iterations **do**
4:         for all capsules $i$ in layer $l$: $c_i \leftarrow softmax(b_i)$.
5:         for all capsules $j$ in layer $(l+1)$: $s_j \leftarrow \sum_i c_i j\hat{u}_{j|i}$.
6:         for all capsules $j$ in layer $(l+1)$: $v_j \leftarrow squash(s_j)$.
7:         for all capsules $i$ in layer $l$ and capsules $j$ in layer $(l+1) : b_{ij} \leftarrow b_{ij} + \hat{u}_{j|i}.v_j$.
8:     **end for** return $V_j$
9: **end procedure**

---

In the initialization phase of dynamic routing calculation, set $b_{ij} = 0$, where $i$ denotes capsule $i$ in layer $l$ and $j$ denotes capsule $j$ in layer $(l+1)$. The internal calculation process of the capsule is shown in Formulas (7)–(11), and the internal calculation diagram is shown in Figure 4.

$$b_1^0 = 0, b_2^0 = 0, T = 5 \tag{7}$$

$$c_1^r, c_2^r = softmax(b_1^{r-1}, b_2^{r-1}) \tag{8}$$

$$s^r = c_1^r u^1 + c_2^r u^2 \tag{9}$$

$$a^r = Squash(s^r) \tag{10}$$

$$b_i^r = b_i^{r-1} + a^r u^i \tag{11}$$

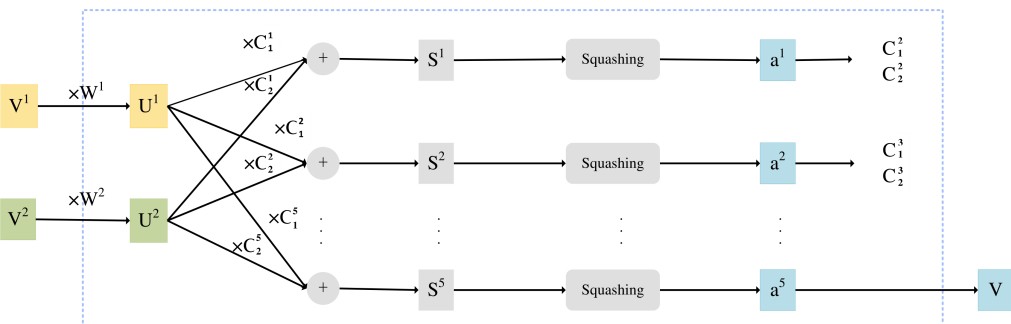

**Figure 4.** Capsule calculation process.

*3.5. XLNet Caps Model*

For the problem we wanted to study, we needed to mine the deep semantic features of the user's speech on the social platform and analyze the user's personality characteristics from these features. For the XLNet-Caps model proposed in this paper, we performed preprocessing operations by inputting the document into XLNet to obtain the corresponding feature map (note that our XLNet intermediate layer had a total of 12 layers). Then, the feature map, which was the output of XLNet, was used as the input to the capsule to calculate the final result. The XLNet-Caps model diagram is shown in Figure 1, and the details of the process are as follows:

1. ReLU conv layer: The output features of XLNet ($4 \times 1024 \times 768$) were processed further, through the dropout layer, the linear layer, and the ReLU activation function, causing the the features to become $8 \times 128 \times 128$ as the input of the capsule. First, a $25618 \times 18$ kernel with a stride of 1 was used to perform the convolution calculation, and the ReLU activation function was used to obtain the output as $111 \times 111 \times 256$, then input it into the primary capsule layer. When setting the kernel size, we experimented with a size from 9–18, and finally, we selected a size of 18 according to the results;

2. Primary capsule layer: The input was a $111 \times 111 \times 256$ tensor, and a $25,618 \times 18$ kernel with a stride of 2 was used for the convolution calculation, then the output was expanded to obtain 565,504 capsules, each capsule having an 8-dimensional vector. When setting the kernel size, we experimented with a size from 9–18, and finally, we selected a size of 18 according to the results;

3. Digit capsule layer: The input was $565,504 \times 8$ capsules, and we first used matrix $W$ to perform a linear change. Note that each $W_{ij}$ was an $8 \times 16$ matrix, and the $\hat{\mu}_{j|i}$ obtained by multiplying by the corresponding $\mu_i$ was a $1 \times 16$ vector. Since there were 10 capsules in the upper layer, the dimension of $W$ was $(8, 16, 565, 504, 10)$. As described in the second part, with the weighted addition of $\hat{\mu}_{j|i}$, the upper layer capsules had $10 \times 16$ dimensions. In the DigitCaps layer, we tested the routing by setting it to 1, 5, and 3, respectively, and finally, we selected 3 according to the results;

4. Fully connected and prediction layer: The output result of the digit capsule layer was input into the fully connected layer to predict the final result.

## 4. Experiments

*4.1. Datasets*

We collected data about personality, including Big Five and MBTI-related content. There are more MBTI data available than Big Five data, so we proposed a method to convert Big Five and MBTI-related content, in order to reasonably use these data. As mentioned earlier, Big Five includes five categories, and these five categories have a corresponding relationship with MBTI, as shown in Table 2.

**Table 2.** Differences and commonalities of the Big Five and MBTI datasets.

| MBTI | Big Five |
|---|---|
| Intuition/sensing | Openness to experience (correlates with N) |
| Feeling/thinking | Agreeableness (correlates with F) |
| Perception/judging | Conscientiousness (correlates with J) |
| Introversion/extraversion | Extraversion (correlates with E |
| Not available in MBTI | Neuroticism |

We conducted many experiments on the two public datasets. The detailed description of each dataset is given as follows:

- Essays. is a scientific gold standard from psychology. James Pennebaker collected stream-of-consciousness essays in a controlled environment, and Laura King labeled the data with the Big Five personality traits. There are 2467 pieces of data in the dataset;
- MBIT: These data were collected through the PersonalityCafe forum (https://www.personalitycafe.com/forum/, accessed on 15 March 2020), which includes many users' text information and MBTI personality types obtained through testing. There are 8675 pieces of data in the MBIT dataset. We used micro-$F_1$ and macro-$F_1$ as our evaluation metrics for hierarchical classification [26];
- Micro-$F_1$ is an $F_1$-score that comprehensively considers factors such as overall accuracy and the recall rate of all tags. Let $TP_t$, $FP_t$, and $FN_t$ denote the true positives, false positives, and false negatives for the *t*-th label in label set *L*, respectively. Then, the micro-averaged $F_1$ is:

$$\mathbf{P} = \frac{\sum_{t \in \mathcal{L}} \mathbf{TP}_t}{\sum_{t \in \mathcal{L}} \mathbf{TP}_t + \mathbf{FP}_t} \tag{12}$$

$$\mathbf{R} = \frac{\sum_{t \in \mathcal{L}} \mathbf{TP}_t}{\sum_{t \in \mathcal{L}} \mathbf{TP}_t + \mathbf{FN}_t} \tag{13}$$

$$\mathbf{R} = \frac{\sum_{t \in \mathcal{L}} \mathbf{TP}_t}{\sum_{t \in \mathcal{L}} \mathbf{TP}_t + \mathbf{FN}_t} \tag{14}$$

$$Micro - \mathbf{F}_1 = \frac{\mathbf{2PR}}{\mathbf{P+R}} \tag{15}$$

- Macro-$F_1$ is another $F_1$-score, which acts in a hierarchy and can evaluate the average $F_1$ of all different category labels. Macro-$F_1$ gives each label the same weight. Formally, the macro-average $F_1$ is:

$$\mathbf{P}_t = \frac{\mathbf{TP}_t}{\mathbf{TP}_t + \mathbf{FP}_t} \tag{16}$$

$$\mathbf{R}_t = \frac{\mathbf{TP}_t}{\mathbf{TP}_t + \mathbf{FN}_t} \tag{17}$$

$$\mathbf{R}_t = \frac{\mathbf{TP}_t}{\mathbf{TP}_t + \mathbf{FN}_t} \tag{18}$$

$$Macro - \mathbf{F}_1 = \frac{1}{|\mathcal{L}|} \sum_{t \in \mathcal{L}} \frac{\mathbf{2P}_t \mathbf{R}_t}{\mathbf{P}_t + \mathbf{R}_t} \tag{19}$$

### 4.2. Hyperparameters

In our experiments, we used the pre-trained XLNet model in the paper [21]. The model consisted of 12 layers; the hidden state size was 768, and the XLNet+capsule model had 175 M parameters. We adjusted the hyperparameters so that the model performed the best. Below are the hyperparameters used in the experiment:

1. The learning rate was set to 0.00002;
2. The value of dropout was set to 0.1;
3. The capsule's convolution kernel was set to $18 \times 18$;
4. The capsule's dynamic routing algorithm was iterated 3 times;
5. The two-class cross-entropy was used as the loss function;
6. The optimizer used was the Adam optimizer.

### 4.3. Comparison of the Methods

#### 4.3.1. Traditional Machine Learning

This type of method first exploits the TF-IDF features from the original text and then inputs them into the classification model, such as logistic regression (LR) and support vector machines (SVMs). Those methods rely heavily on manual feature extraction and basically cannot automatically capture semantic features [27].

### 4.3.2. Neural Network Models

Deep neural networks and representation learning have led to a new ideology for solving the data sparsity problem, and one of the major advantages of neural networks is that they can learn [28]. CNN-based and RNN-based models such as TextCNN and LSTM are mainly used to handle personality classification problems. Hierarchical CNN and hierarchical LSTM are also implemented to explore the validity of temporal features.

### 4.3.3. RNN-Capsule

RNN-capsule is a model [19] for sentiment classification. Its input is first passed through the RNN layer to obtain the hidden layer vector, and then, it enters each capsule to obtain the corresponding classification. RNN-capsule separates different emotion category data and separately inputs them into the corresponding blocks for training so that linguistic information can be extracted from each block after the training has completed. The model structure of RNN-capsule is relatively simple and does not need to use carefully trained word vectors to achieve good classification results.

### 4.3.4. Variations of BERT+Capsule

Since BERT was proposed, it has achieved excellent results in many NLP tasks [29]. The architecture of our proposed model was based on BERT. We performed some experiments on variants of the BERT and BERT-plus-capsule models, such as ALBERT-capsule, RoBERTa-capsule, and XLNet-capsule. The experimental results showed that they could all achieve a certain effect in sentiment classification, and the effect was better than that of the BERT+capsule model. XLNet-capsule performed better than the other two models and achieved the best results.

### 4.4. Experimental Settings

We used a batch size of 128 units for all our experiments. Models were trained on two 1080-Ti GPUs. We used Windows 10 Professional Edition, Python 3.7.4, CUDA 10.0, and PyTorch 1.1.0 as the experimental environment for this experiment. When modeling documents, we set the number of messages to be processed and the maximum length of each message to 40 and 100, respectively. We used the skip-gram algorithm to train the word embedding using the default parameters in Word2Vec. In addition, our model was trained using the Adam optimizer with a learning rate of $2 \times 10^{-5}$.

### 4.5. Performance Comparison

To test the superiority of our proposed method, we compared the proposed XLNet+capsule model with the widely used personality classification methods and the latest methods on the same dataset.

### Comparison to the Baseline Methods

We compared our XLNet-Caps approach to the following baselines.

1. LR : This baseline method uses traditional machine learning methods by preprocessing the text, then uses GloVe to represent the preprocessed text features, and, finally, classifies the text through logistic regression classification. Binary-cross-entropy loss is used as the loss function;
2. SVM: This baseline is similar to LR, but the classification is performed using SVM. The LR and SVM baseline methods represent traditional machine learning methods;
3. TextCNN: This baseline uses the default parameters of TextCNN [12]. The feature vectors are obtained by the text transformed by GloVe. Binary-cross-entropy loss is used as the loss function;
4. LSTM: Long short-term memory (LSTM) is a recurrent neural network and is applied to text classification. Binary-cross-entropy loss is used as the loss function;
5. BERT: We used this pre-trained model without any extension as our baseline. Binary-cross-entropy loss is used as the loss function;

6.　BERT-CNN: This model extracts the features of the text through BERT and then adds a layer of the convolutional neural network to the BERT output to obtain feature information between sentences;

7.　BERT-Caps: This model is similar to BERT-CNN: it is also a multi-model, and a layer of the capsule network is added to the BERT output. We employed the capsule loss to report the results;

8.　RNN-Caps: This model uses a convolutional neural network to replace the BERT model in the BERT-Caps model above. Similarly, the capsule loss is use as the loss function;

9.　ALBERT-Caps: ALBERT is an improvement of BERT that reduces the model parameters. This baseline model uses the ALBERT preprocessing model to extract text features and then sends the obtained features to the capsule network for training, using the capsule loss as the loss function;

10.　RoBERTa-Cap: As ALBERT-Caps, this model replaces ALBERT with RoBERTa, using capsule loss as the loss function.

Table 3 shows the macro-$F_1$ and micro-$F_1$ results of all baseline models and the proposed model. Table 4 shows the recall rate of part of the baselines and the proposed model on each feature and the average of all recall rates. It is obvious from the table that the proposed XLNet-Caps produced better results than all other models.

**Table 3.** The macro-$F_1$ and micro-$F_1$ of all the baselines and the proposed model.

| Method | Macro-$F_1$ | Micro-$F_1$ |
|:---:|:---:|:---:|
| LR | 0.49 | 0.571 |
| SVM | 0.534 | 0.608 |
| TextCNN | 0.57 | 0.464 |
| LSTM | 0.552 | 0.637 |
| BERT | 0.59 | 0.674 |
| BERT-CNN | 0.605 | 0.713 |
| BERT-Caps | 0.641 | 0.672 |
| RNN-Caps | 0.434 | 0.537 |
| ALBERT-Caps | 0.653 | 0.685 |
| RoBERTa-Caps | 0.644 | 0.675 |
| XLNet | 0.592 | 0.595 |
| XLNet-Caps | 0.680 | 0.682 |

Figure 5 visually shows the macro-$F_1$ and micro-$F_1$ performance of each baseline algorithm and XLNet-Caps. It can be seen from the figure that only TextCNN's macro-$F_1$ was greater than its micro-$F_1$. This is because TextCNN is mainly a convolution operation, and convolution in text information mainly considers the information between sentences. The BERT-CNN model, which also uses convolution, is completely different. This is because the text is subjected to BERT for feature extraction, which can fully extract the features in the text. From this figure, it is clear that the best performance index was BERT-CNN's micro-$F_1$, which exceeded 0.7, but its macro-$F_1$ effect was very poor, only about 0.6. Comprehensively looking at the macro-$F_1$ and micro-$F_1$, the best performance was shown by our proposed model, XLNet-Caps.

For the essays dataset, by the comparison in Table 3, in traditional machine learning, SVM was better than LR. SVM had 0.534 and 0.608 for the macro-$F_1$ and micro-$F_1$, respectively, while LR had 0.49 and 0.571 for the macro-$F_1$ and micro-$F_1$, respectively. In contrast, SVM improved the performance of LR by about 3%, which was already considered a big improvement. The reason is that SVM can solve high-dimensional problems and deal with the interaction of nonlinear features. At the same time, in Table 3, compared with the traditional method, all the neural network methods had better performance. The above results showed that the deep neural network model learned richer semantic features. The macro-$F_1$ of LSTM was 0.552, which was smaller than that of TextCNN (0.57), while in the comparison of the micro-$F_1$ indicators, LSTM (0.637) was much higher than TextCNN

(0.464), which was about 17% higher. This is because, in text classification, the recurrent neural network can only model the text sequence in one direction. It can only learn the dependence of a certain position in the sequence on the previous position, but cannot capture the dependence on the subsequent position. This one-way modeling feature does not conform to the bidirectional feature of natural language because the certain position of the natural language will be affected by its context, so the recurrent neural network and its variants cannot obtain the contextual features in the text sequence, but can only obtain a single "above" feature. TextCNN takes into account the relationship between sentences, that is the macro message, so the effect of TextCNN was better than LSTM at the macro-level, and the effect was inferior to LSTM at the micro-level. In the same way, the macro performance of RNN-Caps was weaker than its micro performance.

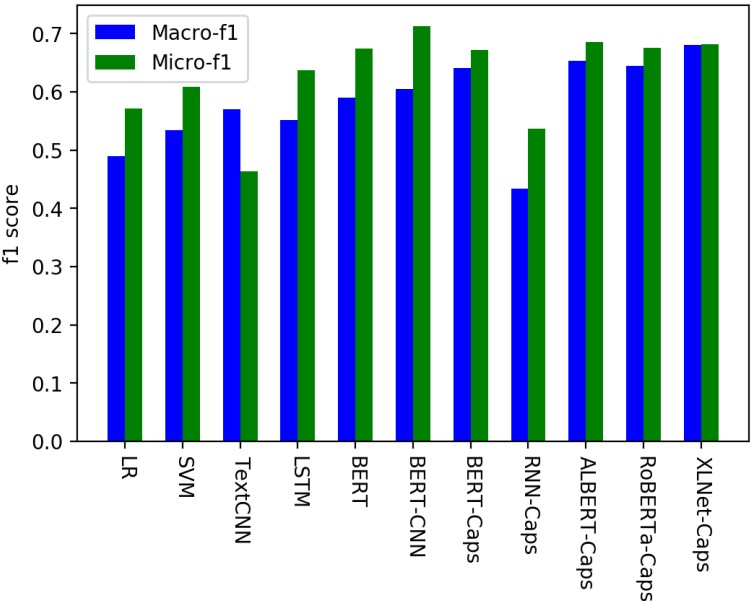

**Figure 5.** The macro-$F_1$ and micro-$F_1$ of all the baselines and the proposed model.

We can see that the BERT model performed better than the traditional machine learning and deep neural network methods. The results showed that the BERT model had powerful pre-training capabilities, which could fully capture the semantic information of each dimension in the text classification task and extract more sufficient features. At the same time, it can be seen from Tables 3 and 4 that the performance of the composite model was better than that of the single model. The BERT+capsule model added the capsule network to the BERT model. First, after passing the BERT model, the emotional semantic features of the text were extracted and then extracted again through the capsule, so the results obtained were better than the results of the single BERT model. The variations of BERT have made some changes based on BERT. The effect of combining the variations of the BERT model and capsule was better than that of BERT-Caps. Compared to BERT-Caps, the macro-$F_1$ index of ALBERT-Caps increased by 1%; RoBERTa-Caps did not show significant improvement; and the macro-$F_1$ index of XLNet-Caps improved by 4%. For the micro-$F_1$ index, that of ALBERT-Caps increased by 1.3%; that of RoBERTa-Caps also increased, but not obviously; and that of XLNet-Caps increased by 1%. In terms of comprehensive indicators, XLNet-Caps had the best effect. It is clear from the comparison of the data that the XLNet-Caps model not only performed best on a single emotional feature, but also performs best on all five emotional features.

**Table 4.** The recall of part of the baselines and the proposed model for every trait.

| Algorithm | Traits | | | | | Average |
| --- | --- | --- | --- | --- | --- | --- |
| | OPN | CON | EXT | AGR | NEU | |
| RNN-Caps | 0.534 | 0.514 | 0.487 | 0.492 | 0.527 | 0.511 |
| LR | 0.606 | 0.617 | 0.721 | 0.541 | 0.71 | 0.639 |
| SVM | 0.646 | 0.653 | 0.76 | 0.606 | 0.693 | 0.672 |
| TextCNN | 0.691 | 0.682 | 0.79 | 0.67 | 0.73 | 0.712 |
| LSTM | 0.682 | 0.674 | 0.781 | 0.673 | 0.713 | 0.704 |
| BERT | 0.70 | 0.694 | 0.791 | 0.673 | 0.72 | 0.715 |
| BERT-CNN | 0.74 | 0.703 | 0.817 | 0.687 | 0.798 | 0.749 |
| BERT-Caps | 0.743 | 0.716 | 0.818 | 0.679 | 0.807 | 0.753 |
| XLNet | 0.723 | 0.726 | 0.802 | 0.676 | 0.799 | 0.768 |
| XLNet-Caps | 0.774 | 0.749 | 0.829 | 0.683 | 0.821 | 0.771 |

*4.6. Evaluation on Sequence Text modeling*

In order to improve the experimental effect of the model and study the trends of emotional changes, we further explored the number of sequence texts in the sequence text modeling. By using the MBTI dataset to process different amounts of sequence texts, we can see in Figure 6 that the number of sequence texts had a great impact on the classification performance. The information sent by people at different times contains different emotions, so a certain number of sequence texts can reflect people's emotional changes and reflect their personality characteristics.

Additionally, people generally post on social media when their emotions change, so it is important to sample their historical data. If the number of sequence texts is not enough to reflect changes in emotions, the text information is invalid.

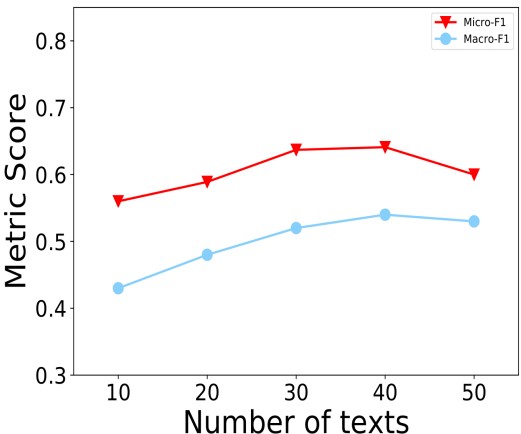

**Figure 6.** Comparing the performance obtained with different numbers of texts.

It was laborious for the model to capture the personality characteristics. If the number of sequence texts is too large, the model may capture redundant information, which would lead to bad performance. As shown in the figure, the model performed the worst on all indicators of the 10 texts, and the sequential text modeling would lose the emotional change information. If we set 40 as the number of texts processed at a time, the indicators of the model would achieve the best results. It was proven that sequential text modeling could effectively constitute the text's semantic representation and emotional change trends. If the number of texts processed at a time was set to 50, the classification accuracy of the model would be poor due to redundant information. This illustrates that the number of different sequence texts in the sequence text modeling process would affect the classification performance. Only after many trials could we find the number of texts with the best performance.

## 5. Conclusions

Personality prediction is a significant domain of future research, and personality traits play an important role in business intelligence, marketing, and psychology. The social behavior of users on social platforms is an excellent source of personality prediction. The Big Five model can help us to identify personality traits through language information. More and more scholars have begun to study the personality traits contained in textual information. Different from other papers, (a) The XLnet-caps model is divided into two modules, one is the encoding module, and the other is the feature extraction module, and some papers only use one module, such as LR, SVM, LSTM, BERT, etc. (b) The encoding module of this article uses XLnet to encode the document, and those articles that also use the two modules use XLnet in the encoding module instead of XLnet, but bag-of-words encoding, RNN, BERT, ALBERT, RoBERTa, etc. (c) Our paper user use capsule as our feature extraction module, while some of other papers use CNN as their feature extraction module. Compared with those papers which also use capsule for feature extraction, the main difference lies in the selection of coding module. In this work, we proposed the XLNet-capsule model, which analyzed the text information of social network users. Full consideration was given to non-continuous semantic information, and the semantic information in the text could be extracted at a deeper level. Determining personality characteristics relying on the Big Five model can be regarded as a "multi-label classification" problem to achieve personality prediction. The experimental evaluation showed that the model could predict the personality characteristics of social network users well.

**Author Contributions:** Project administration, Y.W.; Methodology, J.Z.; Investigation, Q.L.; Writing—original draft, Q.L.; Validation, C.W. and H.Z.; Data curation, J.G.; Formal analysis, J.G. All authors have read and agreed to the published version of the manuscript.

**Funding:** This research received no external funding.

**Conflicts of Interest:** The authors declare no conflict of interest.

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
