# Peer review of "XLNet-Caps: Personality Classification from Textual Posts"

_electronics, doi:10.3390/electronics10111360_

Round 1

Reviewer 1 Report

Please see the following detailed comments:

  1. It would be nice if the authors could include a hyper-parameter selection process for the network layers in 3.5. Such a process may include a manual grid search over a combination of hyper parameters. The selected combination should be the one that outperforms others on most of the datasets.
  2. The authors may want to add the performance of XLNet without a capsule network in Table 3 and 4.
  3. A more detailed description of the datasets is needed, such as the size of training/testing sets for each dataset.
  4. Multiple-authored articles should be correctly referenced in the bibliography. For example, reference 5-9 and some others should have all the authors’ names spelled out.

Author Response

Dear reviewer:
Thank you for your suggestions on the correction of the paper,I replied to your question in the attachment, please check it out, thank you

Reviewer 2 Report

This paper deals with aXLNet-Caps: Personality Classification from Textual Posts. I think this paper well described previous work and is also well organized. However, I would like to point out the following.

  1. It is not clear what is differentiation compare to other papers even though the authors proposed the model, XLNet-Caps.
  2. In section 3.2,  Text Modeling Based on Time Series. what's the mean of the time series? It is not clear. Thus authors should explain it in section 3.2
  3. Please add the following reference if it is related. 
    Isolated Spoken Word Recognition Using One-Dimensional Convolutional Neural Network
    Jihad Anwar Qadir, Abdulbasit K. Al-Talabani, and Hiwa A. Aziz
    International Journal of Fuzzy Logic and Intelligent Systems 2020;20(4):272-277

Author Response

Dear reviewer:
Thank you for your suggestions on the correction of the paper,I replied to your question in the attachment, please check it out, thank you.

Round 2

Reviewer 1 Report

The authors have addressed all the concerns in the revision.

Reviewer 2 Report

I think this paper well revised according to the reviewer's point out. Thus I would like to decide as an "accept"